**Data Availability Statement:** All sequence data are available from NCBI BioProject PRJNA767905

# Rhinovirus/enterovirus was the most common respiratory virus detected in adults with severe acute respiratory infections pre-COVID-19 in Kuala Lumpur, Malaysia

**Yoong Min Chong[1], Yoke Fun Chan[1]\*, Mohamad Fadhil Hadi Jamaluddin[2], M. Shahnaz Hasan[2], Yong Kek Pang[3], Sasheela Ponnampalavanar[3], Sharifah Faridah Syed Omar[3], I-Ching Sam [1]\***

1 Department of Medical Microbiology, Faculty of Medicine, University of Malaya, Kuala Lumpur, Malaysia,
2 Department of Anesthesiology, Faculty of Medicine, University of Malaya, Kuala Lumpur, Malaysia,
3 Department of Medicine, Faculty of Medicine, University of Malaya, Kuala Lumpur, Malaysia

\* chanyf@um.edu.my (YFC); jicsam@ummc.edu.my (ICS)

## Abstract

### Background

Severe acute respiratory infections (SARI) pose a great global burden. The contribution of respiratory viruses to adult SARI is relatively understudied in Asia. We aimed to determine viral aetiology of adult SARI patients in Kuala Lumpur, Malaysia.

### Methods

The prevalence of 20 common (mainly viral) respiratory pathogens, and MERS-CoV, SARS-CoV and 5 bacterial select agents was investigated from May 2017 to October 2019 in 489 SARI adult patients in Kuala Lumpur, Malaysia, using molecular assays (Luminex NxTAG-RPP kit and qPCR assays). Viral metagenomics analysis was performed on 105 negative samples.

### Results

Viral respiratory pathogens were detected by PCR in 279 cases (57.1%), including 10 (2.0%) additional detections by metagenomics analysis. The most detected viruses were rhinovirus/enterovirus (RV/EV) (49.1%) and influenza virus (7.4%). Three melioidosis cases were detected but no SARS-CoV, MERS-CoV or other bacterial select agents. Bacterial/viral co-detections and viral co-detections were found in 44 (9.0%) and 27 (5.5%) cases respectively, mostly involving RV/EV. Independent predictors of critical disease were male gender, chronic lung disease, lack of runny nose and positive blood culture with a significant bacterial pathogen. Asthma and sore throat were associated with increased risk of RV/EV detection, while among RV/EV cases, males and those with neurological disease were at increased risk of critical disease.

(Sequence Read Archive accession numbers SRR16163801-16163905) and PRJNA768949 (Sequence Read Archive accession numbers SRR16214449-SRR16214472), and GenBank (accession numbers OK143237-OK143276).

**Funding:** This study was funded by the Ministry of Education, Malaysia (grant number: FRGS/1/2020/SKK0/UM/02/5; recipients: YFC, ICS); and the Defense Threat Reduction Agency, USA, under Broad Agency Announcement HDTRA1-6 (grant number HDTRA1-17-1-0027; recipients: YFC, MFHJ, MSH, YKP, SP, SFSO, ICS). The funders had no role in study design, data collection and analysis, decision to publish, or preparation of the manuscript.

**Competing interests:** The authors have declared that no competing interests exist.

## Conclusions

Prior to the COVID-19 pandemic, the high prevalence of respiratory viruses in adults with SARI was mainly attributed to RV/EV. Continued surveillance of respiratory virus trends contributes to effective diagnostic, prevention, and treatment strategies.

## Introduction

Globally, respiratory tract infections cause 2.5 million deaths annually [1]. In Malaysia, severe acute respiratory infection (SARI) is the leading cause of morbidity and mortality among children <5 and adults >75 years [2]. As SARIs are commonly caused by viruses, the WHO has launched the Battle against Respiratory Viruses initiative in 2012 [3].

Accurate data on the burden of respiratory viruses is vital for patient management, infection control measures and public health policies. Most studies have been conducted in developed countries and among children. However, the distribution of pathogens varies between countries, and there is limited data available on viral SARIs in adults, particularly in Asia. *Burkholderia pseudomallei*, the bacterial select agent that causes melioidosis, is an important cause of SARIs and is endemic in Malaysia. Other select agents causing SARIs such as MERS-CoV, SARS-CoV, *Bacillus anthracis*, *Yersinia pestis*, *Francisella tularensis* and *Coxiella burnetii* are not routinely tested for, and their prevalence in Malaysia remains unknown.

Metagenomic next-generation sequencing is a sensitive pan-pathogen assay for diagnosis and identification of new or rare pathogens, or those missed by routine diagnostics. No culture, cloning or prior knowledge of pathogens present is required. Metagenomic analysis is of particular interest to Southeast Asian countries, including Malaysia, which are known hotspots for emerging diseases.

We report the viral etiologies in adults admitted with SARI in Kuala Lumpur, Malaysia in the 2 years before COVID-19, using molecular assays and metagenomics. We also evaluated clinical outcomes and predictors of critical SARIs.

## Materials and methods

### Patient enrollment

This study was conducted in University Malaya Medical Centre, a 1600-bed teaching hospital in Kuala Lumpur, Malaysia, from May 2017 to October 2019. Adults aged ≥18 years with community-acquired SARI were prospectively enrolled with written informed consent. A SARI is an acute respiratory infection with fever of ≥38˚C or a history of fever and cough within 10 days and requiring hospitalization [4]. Community-acquired infections are detected within 72 hours of admission. Critical SARI cases require an intensive care unit (ICU), ventilator or inotropic support, or result in death. Age- and sex-matched adults attending outpatient clinics with no respiratory infection in the last month were recruited as controls. A nasopharyngeal swab, oropharyngeal swab, sputum or bronchoalveolar lavage was collected and stored at -80˚C for subsequent molecular analysis. Routine blood cultures were collected and processed with the BacT/ALERT VIRTUO system (bioMérieux, France). The study was approved by the hospital's Medical Research Ethics Committee (no. 20161–2084).

## Nucleic acid extraction and respiratory pathogen detection

Viral and bacterial nucleic acid were extracted using the IndiSpin Pathogen kit (Indical Bioscience, Germany). Twenty respiratory pathogens, including influenza A virus (IAV; A/H1 and A/H3), influenza B virus (IBV), human adenovirus (HAdV), human parainfluenza virus (HPIV, types 1–4), respiratory syncytial virus (RSV type A and B), human metapneumovirus (HMPV), rhinovirus/enterovirus/ (RV/EV), human coronavirus (HCoV-HKU1, -229E, -NL63 and -OC43), human bocavirus (HBoV), *Chlamydophila pneumoniae*, *Mycoplasma pneumoniae* and *Legionella pneumonia* were detected using the NxTAG Respiratory Pathogen Panel (NxTAG RPP) (Luminex, USA). MERS-CoV, SARS-CoV and bacterial select agents (*Burkholderia pseudomallei*, *Bacillus anthracis*, *Yersinia pestis*, *Francisella tularensis* and *Coxiella burnetii*) were tested using published qPCR assays [5–11], which were optimized and validated (S1 Data).

## Viral metagenomics

Samples from selected cases with critical SARIs and/or respiratory comorbidities and negative for all tested pathogens, and healthy controls were subjected to viral metagenomics [12]. Samples were centrifuged at 10,000*g* for 10 min at 4˚C. Supernatants were filtered through 0.45μm ultrafiltration spin-columns (Millipore, Germany) and treated with TURBO DNA-free DNase (Invitrogen, USA) and RNase A (Invitrogen) before incubation for 1 hour at 37˚C. Viral nucleic acid was isolated using QIAamp MinElute Virus Spin kit (QIAGEN, Germany) and treated with 1U/μl DNase I, Amplification Grade (Invitrogen, USA).

Viral nucleic acid was amplified by sequence-independent, single-primer amplification [13] and labelled with tag sequences. First strand cDNA was reverse transcribed using FR26RV-8N primer (GCC GGA GCT CTG CAG ATA TCN NNN NNN N) and Superscript IV First-Strand Synthesis System (Invitrogen). Second strand synthesis was performed using 5U/μl Klenow Fragment (3 → 5' exo-) (NEB, USA).

PCR amplification was performed with primer FR26RV (GCC GGA GCT CTG CAG ATA TC) and AmpliTaq Gold DNA polymerase (Applied Biosystems, USA). The random PCR products were purified with Zymo DNA Clean & Concentrator (Zymo Research, USA). Libraries were prepared using Illumina DNA Prep kit (Illumina, USA) and sequenced on an Illumina NextSeq 500 platform using a NextSeq 500/550 High Output kit v2.5 (300 cycles) (Illumina) with molecular grade water as a non-template control (NTC).

## Bioinformatics analysis

Raw sequencing reads were trimmed to remove adapters and low-quality reads. Host sequences and NTC reads were filtered out before identifying viral pathogens using the Chan Zuckerberg ID portal (https://czid.org/) [14]. A virus was reported if non-overlapping reads from ≥3 distinct genomic regions were identified [15]. Viruses detected in the NTC or known laboratory contaminants were not reported [16]. The reference-based mapping approach was employed to assess level of identity and genome coverage. Raw reads were submitted under NCBI BioProject numbers PRJNA767905 (Sequence Read Archive SRR16163801-16163905) and PRJNA768949 (SRR16214449-SRR16214472).

## Genotyping of rhinovirus/enterovirus

Reverse transcription semi-nested PCR was used to genotype rhinoviruses (5' untranslated region and viral protein 4/viral protein 2 (VP4/VP2) transition region) and enteroviruses

(VP1) (S1 Table) [17, 18]. Sanger sequencing was performed and sequences deposited in Gen-Bank (accession numbers OK143237-OK143276).

### Phylogenetic analysis of rhinoviruses

Study sequences were aligned with publicly available complete rhinovirus genomes and Malaysian rhinovirus sequences using MAFFT in Geneious Prime 2020 (Biomatters, New Zealand) with default parameters. A phylogenetic tree based on 432 bp of VP4/VP2 was constructed with IQ-TREE v2.1.2 using the GTR+F+G4 model with 1000 ultrafast bootstrap replicates and visualized with FigTree v1.4.4 [19].

### Data analysis

Multivariate analysis was performed to determine independent predictors of critical disease. As RV/EV was the most frequently detected virus, we also determined factors associated with RV/EV detection and critical RV/EV cases. Potential predictors were tested with univariate logistic regression, generating odds ratios (OR) and 95% confidence intervals (CI). Those with p-values ≤0.2 were included in multivariate analysis using stepwise selection and the likelihood ratio test. Predictors with an adjusted OR with two-sided p≤ 0.05 were considered significant. The final model was assessed with the Hosmer and Lemeshow goodness-of-fit test and the area under the curve (AUC) of the receiver operating characteristic (ROC) curve. IBM SPSS version 23 (IBM, USA) was used.

## Results

### Study population

We enrolled 489 SARI patients; 53.4% were female, the median age was 66 years (range, 19–100), and 21.7% (106/489) had critical SARI (Table 1). Most patients (87.5%) had comorbidities, led by hypertension (58.7%), chronic lung disease (45.8%) and diabetes (42.7%). There were 24 healthy control subjects, with 54.2% females, median age of 66 years (range, 27–83), and 79.2% had underlying diseases. Chronic lung disease (50%) was the most common, followed by hypertension (37.5%) and diabetes (29.2%).

### Detection of respiratory pathogens using molecular assays

From the 489 SARI patients, 421 (86.1%) oropharyngeal swabs, 55 (11.2%) nasopharyngeal swabs, 12 (2.5%) sputum samples and 1 (0.2%) bronchoalveolar lavage specimen were obtained. A total of 271 (55.4%) patients had detectable respiratory pathogens using the molecular assays alone (Table 2). The most common identified virus was RV/EV (48.3%; 236/489), followed by influenza virus (6.1%; 30/489) and others at <2%, such as HMPV, HPIV, RSV, HCoV-OC43, HAdV and HBoV. *B. pseudomallei* was detected in three patients (including one co-detection with HCoV-OC43), and confirmed by positive blood cultures. No case was positive for MERS-CoV, SARS-CoV and other bacterial select agents. Of the 26 viral co-detection cases with ≥2 viruses, RV/EV (96.2%; 25/26) was most frequently identified, especially in combination with influenza virus (57.7%; 15/26) and RSV (11.5%; 3/26). Two (8.3%) of the 24 healthy subjects were positive for any pathogen, and both were RV/EV.

Blood cultures were collected in 453 (92.6%) of the cases, of which 33 (7.3%) yielded bacteria considered to be clinical significant (Table 1). Of the total 489 cases, 18 (3.7%) had a bacterial/viral co-detection, that is a significant blood culture isolate and a detectable respiratory virus, and 15 of these had RV/EV.

**Table 1. Risk variables associated with critical disease and critical rhinovirus/enterovirus disease.**

| Variables | Total cases, n = 489, no. (%) | Critical disease,[†] n = 106, no. (%) | Non-critical disease, n = 383, no. (%) | Univariate analysis OR (CI 95%) | p-value | Multivariate analysis OR (CI 95%) | p-value | Total RV/EV cases, n = 240, no. (%) | Critical RV/EV,[†] n = 52, no. (%) | Non-critical RV/EV, n = 188, no. (%) | Univariate analysis OR (CI 95%) | p-value | Multivariate analysis OR (CI 95%) | p-value |
|---|---|---|---|---|---|---|---|---|---|---|---|---|---|---|
| **Age**, mean (standard deviation) | 63.7 (16.1) | 64.4 (15.8) | 63.5 (16.2) | 1.003 (0.99–1.017) | 0.64 | | | 62.2 (17.1) | 63.5 (16.9) | 61.9 (17.2) | 1.006 (0.987–1.024) | 0.55 | | |
| **Gender** | | | | | | | | | | | | | | |
| Female | 261 (53.4%) | 43 (40.6%) | 218 (56.9%) | Ref | | Ref | | 129 (53.8%) | 17 (32.7%) | 112 (59.6%) | Ref | | Ref | |
| Male | 228 (46.6%) | 63 (59.4%) | 165 (43.1%) | 1.936 (1.250–2.997) | 0.003* | 1.870 (1.174–2.979) | 0.01* | 111 (46.3%) | 35 (67.3%) | 76 (40.4%) | 3.034 (1.586–5.803) | 0.001* | 2.959 (1.535–5.704) | 0.001* |
| **Underlying diseases** | | | | | | | | | | | | | | |
| Asthma | 156 (31.9%) | 30 (28.3%) | 126 (32.9%) | 0.805 (0.502–1.292) | 0.37 | | | 89 (37.1%) | 19 (36.5%) | 70 (37.2%) | 0.971 (0.513–1.836) | 0.93 | | |
| Diabetes | 209 (42.7%) | 43 (40.6%) | 166 (43.3%) | 0.892 (0.576–1.392) | 0.61 | | | 104 (43.4%) | 23 (44.2%) | 81 (43.1%) | 1.048 (0.564–1.945) | 0.88 | | |
| Hypertension | 287 (58.7%) | 63 (59.4%) | 224 (58.5%) | 1.040 (0.671–1.611) | 0.86 | | | 137 (57.1%) | 32 (61.5%) | 105 (55.9%) | 1.265 (0.675–2.371) | 0.46 | | |
| Chronic lung disease | 224 (45.8%) | 56 (52.8%) | 168 (43.9%) | 1.433 (0.931–2.207) | 0.10 | 2.111 (1.302–3.422) | 0.002* | 120 (50.0%) | 29 (55.8%) | 91 (48.4%) | 1.344 (0.725–2.492) | 0.35 | | |
| Chronic cardiovascular disease | 108 (22.1%) | 27 (25.5%) | 81 (21.1%) | 1.274 (0.772–2.103) | 0.34 | | | 48 (20.0%) | 13 (25.0%) | 35 (18.6%) | 1.457 (0.704–3.015) | 0.31 | | |
| Chronic kidney disease | 57 (11.7%) | 18 (17.0%) | 39 (10.2%) | 1.804 (0.985–3.306) | 0.06 | | NS | 29 (12.1%) | 10 (19.2%) | 19 (10.1%) | 2.118 (0.917–4.891) | 0.08 | | NS |
| Chronic liver disease | 7 (1.4%) | 2 (1.9%) | 5 (1.3%) | 1.454 (0.278–7.601) | 0.66 | | | 1 (0.4%) | 0 | 1 (0.5%) | - | | | |
| Neurological disease | 46 (9.4%) | 13 (12.3%) | 33 (8.6%) | 1.483 (0.750–2.930) | 0.26 | | | 23 (9.6%) | 10 (19.2%) | 13 (6.9%) | 3.205 (1.315–7.809) | 0.01* | 3.029 (1.207–7.602) | 0.02* |
| Cancer/immunosuppression | 25 (5.1%) | 6 (5.7%) | 19 (5.0%) | 1.149 (0.447–2.955) | 0.77 | | | 8 (3.3%) | 3 (5.8%) | 5 (2.7%) | 2.241 (0.517–9.704) | 0.28 | | |
| **Clinical symptoms** | | | | | | | | | | | | | | |
| Runny nose | 114 (23.3%) | 14 (13.2%) | 100 (26.1%) | 0.431 (0.235–0.790) | 0.01* | 0.362 (0.185–0.707) | 0.003* | 61 (25.4%) | 9 (17.3%) | 52 (27.7%) | 0.547 (0.249–1.202) | 0.13 | | NS |
| Sore throat | 108 (22.1%) | 13 (12.3%) | 95 (24.8%) | 0.424 (0.227–0.794) | 0.01* | | NS | 65 (27.1%) | 10 (19.2%) | 55 (29.3%) | 0.576 (0.270–1.228) | 0.15 | | NS |
| Sputum | 425 (86.9%) | 90 (84.9%) | 335 (87.5%) | 0.806 (0.437–1.486) | 0.49 | | | 204 (85.0%) | 46 (88.5%) | 158 (84.0%) | 1.456 (0.571–3.712) | 0.43 | | |
| **Blood culture positive**[††] | | | | | | | | | | | | | | |
| Yes | 33 (7.3%) | 12 (12.1%) | 21 (5.9%) | 2.187 (1.036–4.619) | 0.04* | 3.000 (1.352–6.658) | 0.01* | 15 (7.0%) | 4 (8.9%) | 11 (6.5%) | 1.410 (0.427–4.658) | 0.57 | | |

*(Continued)*

**Table 1.** (Continued)

| Variables | Total cases, n = 489, no. (%) | Critical disease,[†] n = 106, no. (%) | Non-critical disease, n = 383, no. (%) | Univariate analysis OR (CI 95%) | p-value | Multivariate analysis OR (CI 95%) | p-value | Total RV/EV cases, n = 240, no. (%) | Critical RV/EV,[†] n = 52, no. (%) | Non-critical RV/EV, n = 188, no. (%) | Univariate analysis OR (CI 95%) | p-value | Multivariate analysis OR (CI 95%) | p-value |
|---|---|---|---|---|---|---|---|---|---|---|---|---|---|---|
| No | 420 (92.7%) | 87 (87.9%) | 333 (94.1%) | Ref | | | | 200 (93.0%) | 41 (91.1%) | 159 (93.5%) | Ref | | | |
| Not tested | 36 | 7 | 29 | Excluded | | | | 25 | 7 | 18 | Excluded | | | |
| **Detection of any respiratory virus** | 279 (57.1%) | 62 (58.5%) | 217 (56.7%) | 1.078 (0.697–1.667) | 0.74 | | | | | | Not done | | | |
| **Detection of rhinovirus/ enterovirus** | 240 (49.1%) | 52 (49.1%) | 188 (49.1%) | 0.999 (0.650–1.536) | 0.99 | | | | | | Not done | | | |
| **Detection of influenza virus** | 36 (7.4%) | 7 (6.6%) | 29 (7.6%) | 0.863 (0.367–2.029) | 0.74 | | | 16 (6.7%) | 2 (3.8%) | 14 (7.4%) | 0.497 (0.109–2.261) | 0.37 | | |
| **Co-detection of ≥2 pathogens** | 44 (9.0%) | 11 (10.4%) | 33 (8.6%) | 1.228 (0.598–2.521) | 0.58 | | | 41 (17.1%) | 10 (19.2%) | 31 (16.5%) | 1.206 (0.547–2.657) | 0.64 | | |

[*] Significant p<0.05.

[†] Critical cases are those admitted to ICU, requiring ventilation or inotropes, or resulting in death.

[††] Excludes 36 patients who did not have blood cultures collected.

Significant pathogens were: *Klebsiella pneumoniae* (7), *Staphylococcus aureus* (4), *Streptococcus pneumoniae* (4), *Escherichia coli* (4), *Burkholderia pseudomallei* (3), *Salmonella* species (2), *Proteus mirabilis* (1), *Enterococcus faecalis* (1), *Enterobacter cloacae* (1), *Moraxella* sp. (1), *Prevotella bivia* (1), *Streptococcus dysgalactiae* (1) and polymicrobial cultures (3)

OR, odds ratio; CI, confidence intervals; Ref, parameter of reference; NS, non-significant.

## Viral metagenomics analysis

Nasopharyngeal swab samples from 24 healthy controls underwent viral metagenomics analysis. Raw reads per sample ranged from 12,595,504 to 17,421,763, and after human and contamination reads were filtered, 1.5% were viral reads. One control had a detectable human virus, torque teno virus (Table 3).

Among the 218 samples with negative molecular assays, 105 (48.2%) with critical SARI and/or respiratory comorbidities were selected for viral metagenomics analysis. These comprised 10 nasopharyngeal and 95 oropharyngeal swabs. Raw reads ranged from 9,287,586 to 17,808,836, and 3.7% were viral reads. Sixteen (15.2%) samples had specific human viral reads, of which 10 had respiratory virus pathogens, comprising rhinovirus A (3), IAV/H3 (4), IBV (1), HCoV-OC43 (1) and co-detection with IBV and rhinovirus A (1). Additionally, human papillomaviruses (3), human gammaherpes virus 4 or Epstein-Barr virus (2), torque teno virus (1) and SEN virus (1) were identified but were not considered respiratory pathogens. The addition of viral metagenomics to the molecular assays increased the respiratory pathogen detection rate from 55.4% (271/489) to 57.5% (281/489).

## Seasonal variations of respiratory viruses

The two most commonly detected respiratory viruses, enterovirus/rhinovirus and influenza virus, were detected across the study period with no seasonality noted (Fig 1).

**Table 2. Respiratory pathogens detected by molecular assays and next-generation sequencing in clinical samples.**

| Respiratory pathogens | No. of detections (%) | | |
|---|---|---|---|
| | Molecular assays only (489 samples) | Viral metagenomics only (105 samples negative by molecular assays) | Combined (489 samples) |
| **Rhinovirus/enterovirus (RV/EV)** | 236 | 4 | 240 (49.1%) |
| **Influenza virus** | 30 | 6 | 36 (7.4%) |
| A/H1 | 8 | 0 | 8 (1.6%) |
| A/H3 | 13 | 4 | 17 (3.5%) |
| A/untyped | 4 | 0 | 4 (0.8%) |
| B | 5 | 2 | 7 (1.4%) |
| **Human metapneumovirus (HMPV)** | 8 | 0 | 8 (1.6%) |
| **Human parainfluenza virus (HPIV)** | 10 | 0 | 10 (2.0%) |
| HPIV-3 | 9 | 0 | 9 (1.8%) |
| HPIV-4 | 1 | 0 | 1 (0.2%) |
| **Respiratory syncytial virus (RSV)** | 5 | 0 | 5 (1.0%) |
| RSV-A | 2 | 0 | 2 (0.4%) |
| RSV-B | 3 | 0 | 3 (0.6%) |
| **Coronavirus OC-43 (HCoV-OC43)** | 5 | 0 | 5 (1.0%) |
| **Human adenovirus (HAdV)** | 1 | 0 | 1 (0.2%) |
| **Human bocavirus (HBoV)** | 1 | 0 | 1 (0.2%) |
| **Burkholderia pseudomallei** | 3 | 0 | 3 (0.6%) |
| **Co-detection of viruses*** | 26 | 1 | 27 (5.5%) |
| **Positive** | 271 | 10 | 281 (57.5%) |
| **Negative** | 218 | 95 | 208 (42.5%) |
| **Total** | 489 | 105 | 489 |

*Co-detection cases including RV/EV + A/H1 (2), RV/EV + A/H3 (7), RV/EV + A/untyped (1), RV/EV + influenza B virus (6), RV/EV + RSV-A (1), RV/EV + RSV-B (2), RV/EV + HCoV-OC43 (2), RV/EV + HPIV-3 (2), RV/EV + HMPV (2), RV/EV + HAdV (1), and HPIV-3 + HPIV-4 (1).

## Predictors of critical disease and RV/EV detection

With critical disease as the outcome (Table 1), the independent predictors were male gender (adjusted OR (95% CI), 1.870 (1.174–2.979); p = 0.01), chronic lung disease (OR 2.111 (1.302–3.422); p = 0.002), lack of runny nose (OR 0.362 (0.185–0.707); p = 0.003) and positive blood culture (OR 3.000 (1.352–6.658); p = 0.01). Detection of any respiratory virus, RV/EV, or influenza virus did not predict severity. This model had satisfactory fit and discrimination (Hosmer-Lemeshow goodness-of-fit, $\chi^2$ = 7.03, p = 0.32; ROC AUC = 0.66 (0.61–0.72), p<0.001).

After multivariate analysis using RV/EV detection as the outcome (Table 4), the independent predictors were asthma (OR, 1.508 (1.022–2.224); p = 0.04) and sore throat (OR 1.674 (1.078–2.600); p = 0.02). This model had satisfactory fit and discrimination (goodness-of-fit, $\chi^2$ = 1.34, p = 0.51; ROC AUC = 0.57 (0.52–0.62), p = 0.01).

Predictors for critical disease among the 240 RV/EV cases (Table 1) were male gender (OR 2.959 (1.535–5.704); p = 0.001) and underlying neurological disease (OR 3.029 (1.207–7.602); p = 0.002). This model also had satisfactory fit and discrimination (goodness-of-fit, $\chi^2$ = 0.001, p = 0.97; ROC AUC = 0.66 (0.58–0.75), p<0.001).

**Table 3. Human viruses detected by viral metagenomics analysis.**

| No. | Patient group | Virus detected | Contig count | % covered | Average depth | No. of unique viral reads | Breadth | Viral reads | Total raw reads | Total clean reads | % viral reads |
|---|---|---|---|---|---|---|---|---|---|---|---|
| 1 | SARI | HRV-A40 | 3 contigs | 21.4 | 3.4 | 164 | 1,525 | 222 | 13,972,956 | 3,271,359 | 0.01 |
| 2 | SARI | HRV-A1B | 3 contigs | 88.9 | 1,722.3 | 82,654 | 6,302 | 87,910 | 9,321,610 | 2,056,907 | 4.3 |
| 3 | SARI | Influenza A virus (H3N2) | 8 contigs | 96.3 | 596.9 | 7,774 | 1,801 | 50,796 | 14,507,936 | 282,903 | 17.9 |
| | | Segment 1 (PB2) | 1 contig | 98.9 | 183.6 | 2,898 | 2,292 | | | | |
| | | Segment 2 (PB1) | 1 contig | 97.7 | 662.9 | 10,833 | 2,263 | | | | |
| | | Segment 3 (PA) | 1 contig | 99.5 | 884.0 | 13,454 | 2,197 | | | | |
| | | Segment 4 (HA) | 1 contig | 95.2 | 343.5 | 4,057 | 1,653 | | | | |
| | | Segment 5 (NP) | 1 contig | 91.6 | 646.6 | 6,793 | 1,411 | | | | |
| | | Segment 6 (NA) | 1 contig | 96.7 | 1,273.0 | 12,495 | 1,394 | | | | |
| | | Segment 7 (M1 and M2) | 1 contig | 93.1 | 119.2 | 820 | 933 | | | | |
| | | Segment 8 (NS1 and NEP) | 1 contig | 97.7 | 662.7 | 10,841 | 2,263 | | | | |
| 4 | SARI | Human papillomavirus type 105† | 3 contigs | 43.9 | 2.6 | 169 | 3,362 | 3,112 | 12,982,548 | 129,641 | 2.4 |
| 5 | SARI | HRV-A82 | 4 contigs | 88.2 | 2010.5 | 95,405 | 6,134 | 98,334 | 11,770,096 | 2,903,706 | 3.4 |
| 6 | SARI | Influenza B virus (segment 4 (HA)) | 2 contigs | 66.1 | 33.3 | 392 | 1,160 | 5,546 | 13,097,318 | 2,865,454 | 0.2 |
| | | HRV-A82 | 3 contigs | 9.9 | 79.3 | 689 | 3,723 | | | | |
| | | Human papillomavirus type 38* | 3 contigs | 13.9 | 6.9 | 352 | 1,029 | | | | |
| 7 | SARI | Influenza A virus (H3N2) (segment 1 (PB2)) | 1 contig | 17.3 | 15.8 | 247 | 397 | 1,268 | 14,215,064 | 3,156,013 | 0.04 |
| 8 | SARI | Influenza A virus (H3N2) (segment 2 (PB1)) | 1 contig | 12.8 | 7.5 | 122 | 297 | 1,050 | 13,807,760 | 1,919,628 | 0.1 |
| 9 | SARI | SEN virus* | 7 contigs | 90.3 | 780.1 | 26,754 | 3,466 | 49,944 | 13,943,622 | 351,472 | 14.2 |
| 10 | SARI | Human gammaherpesvirus 4* | 79 contigs | 42.7 | 29.7 | 41,542 | 73,412 | 179,056 | 13,870,150 | 1,872,558 | 9.6 |
| 11 | SARI | Torque teno virus 1* | 3 contigs | 61.5 | 23.1 | 787 | 2,370 | 27,664 | 13,501,382 | 3,146,954 | 0.9 |
| 12 | SARI | Human coronavirus OC43 | 3 contigs | 10.8 | 1.5 | 354 | 3,301 | 427 | 13,417,438 | 424,602 | 0.1 |
| 13 | SARI | Influenza B virus (segment 6 (NA)) | 1 contig | 34.1 | 4.5 | 60 | 534 | 52,431 | 13,269,840 | 4,205,950 | 1.3 |
| 14 | SARI | Influenza A virus (H3N2) | 6 contigs | 35.0 | 24.2 | 386 | 687 | 1,590 | 13,173,080 | 372,006 | 0.4 |
| | | Segment 1 (PB2) | 1 contig | 20.4 | 1.9 | 34 | 473 | | | | |
| | | Segment 2 (PB1) | 1 contig | 23.5 | 8.3 | 164 | 545 | | | | |
| | | Segment 3 (PA) | 1 contig | 58.1 | 78.1 | 1,328 | 1,283 | | | | |
| | | Segment 4 (HA) | 1 contig | 33.0 | 2.7 | 34 | 562 | | | | |
| | | Segment 6 (NA) | 1 contig | 39.9 | 29.9 | 372 | 573 | | | | |
| 15 | SARI | Human papillomavirus type 20* | 3 contigs | 49.2 | 5.0 | 313 | 3,808 | 15,180 | 13,029,066 | 2,704,987 | 0.6 |
| 16 | SARI | Human gammaherpesvirus 4* | 80 contigs | 30.4 | 14.6 | 9,430 | 55,973 | 18,860 | 13,788,806 | 1,423,282 | 1.3 |
| 17 | Healthy control | Torque teno virus 16* | 6 contigs | 95.5 | 67.1 | 2,850 | 2,914 | 19,372 | 12,659,428 | 1,029,926 | 1.9 |

* Not considered respiratory pathogens in this study.

## Genetic characterization of rhinovirus/enterovirus

Only 49/240 (20.4%) RV/EV positive samples could be sequenced. The most prevalent was RV-A (59.2%; 29/49), then RV-C (26.5%; 13/49), and 1–2 cases each of RV-B, EV-C104, coxsackievirus B3 and EV-D68. The genetic variability of RV was very wide, positioning in

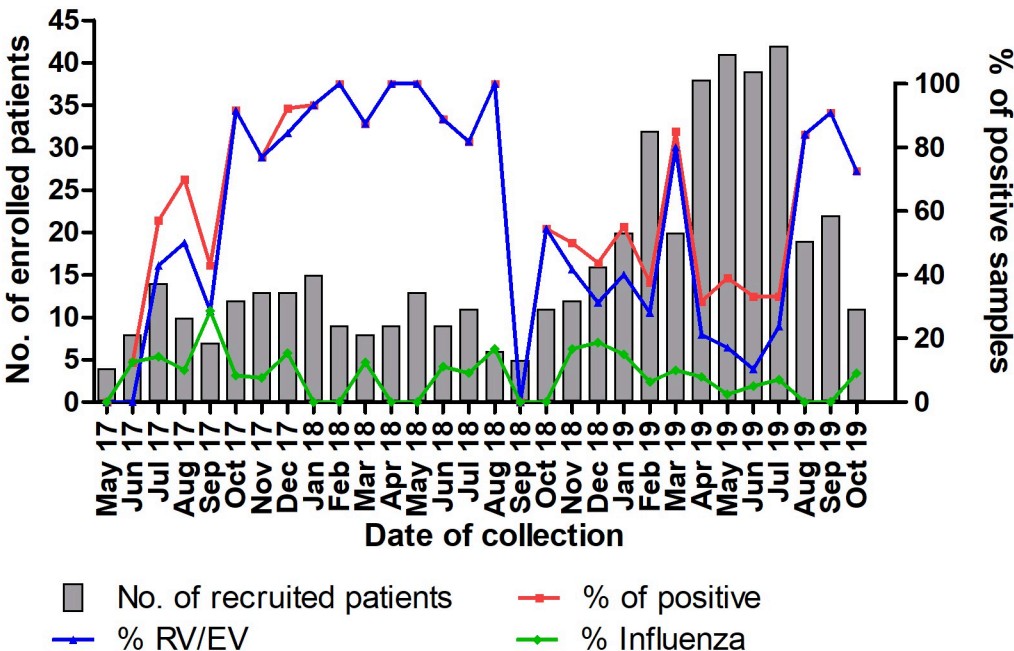

**Fig 1. Seasonal distribution of rhinovirus/enterovirus and influenza virus in SARI adult patients.**

different phylogenetic clusters (Fig 2). We observed 19 RV-A, 11 RV-C, and 2 RV-B genotypes, and some have been previously reported in Malaysia.

## Discussion

There is limited data on respiratory viruses in adult SARI patients in Malaysia, where most previous studies have focused on children [20, 21]. We used a comprehensive panel of molecular assays and viral metagenomics to identify respiratory pathogens in 57.5% of adult SARI patients, with RV/EV (49.1%) and influenza virus (7.4%) the most frequently detected.

RV/EV is the commonest detected virus in adult SARI patients [22, 23]. RV/EV was detected here almost every month, though seasonality in tropical countries is unclear [24]. Our finding that RV-A (59.2%) predominated over RV-C (26.5%) and RV-B (4.1%) is consistent with worldwide studies (RV-A, 35.9–67.7%; RV-C, 23–59.3%; RV-B, 1.5–13%) [25, 26]. However, with a low genotyping success rate, we could not find associations between RV genotypes and critical clinical outcomes. Furthermore, as discussed later, there is inconsistent evidence for the clinical significance of RV/EV detection in SARI.

The influenza positivity rate in this study is within the range (5–14%) reported in tropical countries [27, 28]. Influenza is typically present year-round in tropical countries, with no consistent seasonal peaks [27–29], and remains an underappreciated contributor to respiratory morbidity. RSV is the predominant respiratory virus affecting young children worldwide but not in adult patients, who have detection rates ranging from 0% to 3.9% [22, 23]. We found that RSV was only of minor importance in Malaysian adults with SARI, being detected in only 1% of cases. Apart from SARS-CoV-2, the limited resources for molecular diagnostics in most hospitals here should therefore be focused on RV/EV and influenza virus for adults.

*B. pseudomallei* is endemic in Malaysia, although relatively uncommon in urban Kuala Lumpur [30]. Three cases were identified, two required ICU, and one died. *B. pseudomallei*

**Table 4. Risk variables associated with rhinovirus/enterovirus detection.**

| Variables | Total, n = 489 no. (%) | RV/EV positive[†] n = 240, no. (%) | RV/EV negative n = 249, no. (%) | Univariate analysis | | Multivariate analysis | |
|---|---|---|---|---|---|---|---|
| | | | | OR (CI 95%) | p-value | OR (CI 95%) | p-value |
| **Age**, mean (standard deviation) | 63.7 (16.1) | 62.2 (17.1) | 65.2 (14.9) | 0.989 (0.978–1.000) | 0.01* | | NS |
| **Gender** | | | | | | | |
| Female | 261 (53.4%) | 129 (53.8%) | 132 (53.0%) | Ref | | | |
| Male | 228 (46.6%) | 111 (46.3%) | 117 (47.0%) | 0.971 (0.680–1.385) | 0.87 | | |
| **Underlying diseases** | | | | | | | |
| Asthma | 156 (31.9%) | 89 (37.1%) | 67 (26.9%) | 1.601 (1.091–2.349) | 0.02* | 1.508 (1.022–2.224) | 0.04* |
| Diabetes | 209 (42.7%) | 105 (43.4%) | 104 (42.1%) | 1.049 (0.733–1.501) | 0.79 | | |
| Hypertension | 287 (58.7%) | 137 (57.1%) | 150 (60.2%) | 0.878 (0.612–1.259) | 0.48 | | |
| Chronic lung disease | 224 (45.8%) | 120 (50.0%) | 104 (41.8%) | 1.394 (0.976–1.992) | 0.07 | | NS |
| Chronic cardiovascular disease | 108 (22.1%) | 48 (20.0%) | 60 (24.1%) | 0.788 (0.513–1.210) | 0.28 | | |
| Chronic kidney disease | 57 (11.7%) | 29 (12.1%) | 28 (11.2%) | 1.085 (0.624–1.885) | 0.77 | | |
| Chronic liver disease | 7 (1.4%) | 1 (0.4%) | 6 (2.4%) | 0.169 (0.020–1.418) | 0.06 | | NS |
| Neurological disease | 46 (9.4%) | 23 (9.6%) | 23 (9.3%) | 1.041 (0.567–1.911) | 0.89 | | |
| Cancer/immunosuppression | 25 (5.1%) | 8 (3.3%) | 17 (6.8%) | 0.471 (0.199–1.112) | 0.08 | | NS |
| **Critical disease[††]** | | | | | | | |
| Yes | 106 (21.7%) | 52 (21.7%) | 54 (21.7%) | 0.999 (0.650–1.536) | 0.99 | | |
| No | 383 (78.3%) | 188 (78.3%) | 195 (78.3%) | Ref | | | |
| **Hospitalized in ICU** | | | | | | | |
| Yes | 23 (4.7%) | 12 (5.0%) | 11 (4.4%) | 1.139 (0.493–2.633) | 0.76 | | |
| No | 466 (95.3%) | 228 (95.0%) | 238 (95.6%) | Ref | | | |
| **Ventilation requirement** | | | | | | | |
| Yes | 98 (20.0%) | 48 (20.0%) | 50 (20.1%) | 0.995 (0.639–1.549) | 0.98 | | |
| No | 391 (80.0%) | 192 (80.0%) | 199 (79.9%) | Ref | | | |
| **Death** | 24 (4.9%) | 12 (5.0%) | 12 (4.8%) | 1.039 (0.458–2.361) | 0.93 | | |
| **Clinical symptoms** | | | | | | | |
| Runny nose | 114 (23.3%) | 61 (25.4%) | 53 (21.3%) | 1.260 (0.828–1.918) | 0.28 | | |
| Sore throat | 108 (22.1%) | 65 (27.1%) | 43 (17.3%) | 1.779 (1.152–2.749) | 0.01* | 1.674 (1.078–2.600) | 0.02* |
| Sputum | 425 (86.9%) | 204 (85.0%) | 221 (88.8%) | 0.718 (0.423–1.219) | 0.22 | | |
| **Blood culture positive[†††]** | | | | | | | |
| Yes | 33 (7.3%) | 15 (7.0%) | 18 (7.6%) | 0.917 (0.450–1.867) | 0.81 | | |
| No | 420 (92.7%) | 200 (93.0%) | 220 (92.4%) | Ref | | | |

(*Continued*)

**Table 4.** (Continued)

| Variables | Total, n = 489 no. (%) | RV/EV positive[†] n = 240, no. (%) | RV/EV negative n = 249, no. (%) | Univariate analysis | | Multivariate analysis | |
|---|---|---|---|---|---|---|---|
| | | | | OR (CI 95%) | p-value | OR (CI 95%) | p-value |
| Not tested | 36 | 25 | 11 | Excluded | | | |

[*] Significant p<0.05.

[†] Including 27 co-detections with rhinovirus/enterovirus.

[††] Critical cases are those admitted to ICU, requiring ventilation or inotropes, or resulting in death.

[†††] Excludes 36 patients who did not have blood cultures collected.

OR, odds ratio; CI, confidence intervals; Ref, parameter of reference; NS, non-significant.

has a high fatality rate (10–50%), and molecular assays improve detection for earlier treatment with appropriate antibiotics [31].

Rhinoviruses are frequently detected throughout life and reported in 10–35% of asymptomatic subjects, which may represent true infection or remnants of resolved infection, making it challenging to determine clinical significance [32–34]. Conversely, detections of influenza, RSV, AdV and HMPV are rare in asymptomatic individuals and are highly likely to be clinically important [35, 36]. Meta-analysis of ARI in adults ≥65 years showed strong evidence of causality for PIV, RV, and CoV, but not BoV [37]. While the growing availability of affordable multiplex respiratory panels is welcome, detection of certain pathogens still requires clinical correlation.

Critical SARI was associated with male gender, chronic lung disease, and lack of runny nose, while critical RV/EV was associated with male gender and underlying neurological disease (mostly past strokes). These findings can be used to identify patients needing closer monitoring or hospitalization. Asthma and sore throat were independent predictors of RV/EV infection. Sore throats are more common in RV/EV patients, and rhinoviruses are associated with asthma exacerbations in children and adults [38, 39].

Metagenomics detected 10 (9.5%) additional viral pathogens in 105 samples tested, excluding EBV, torque teno virus and betapapillomaviruses which commonly colonise healthy populations. The detected pathogens were missed by molecular assays, likely due to low viral load and/or primer mismatches. Nevertheless, a significant number of cases remained without an identifiable causative agent. Sampling method, sample preparation, sequencing depth and bioinformatics techniques all affect the sensitivity of metagenomics analysis [40]. Unclear additional clinical yield, high cost and long turnaround time are further barriers to use of metagenomics analysis as routine diagnostics.

This study had limitations. It involved a single hospital, and only 105/218 (48.2%) negative samples underwent viral metagenomics analysis. Broader and more extensive surveillance studies are needed for more nationally representative data. We focused mainly on viral pathogens and did not include analysis of bacterial and fungal cultures, as the clinical significance of these in respiratory samples can be difficult to interpret. This study was conducted before the COVID-19 pandemic, which was associated with declines in other respiratory viruses globally, including at our centre [41, 42]. Nevertheless, it provides important baseline data of circulating respiratory viruses in a tropical country. Continued surveillance is important to determine the epidemiological patterns of respiratory viruses, particularly as other viruses will re-emerge post-pandemic [43], and provide data for public health policies and appropriate resources for diagnostics, treatment, and vaccines.

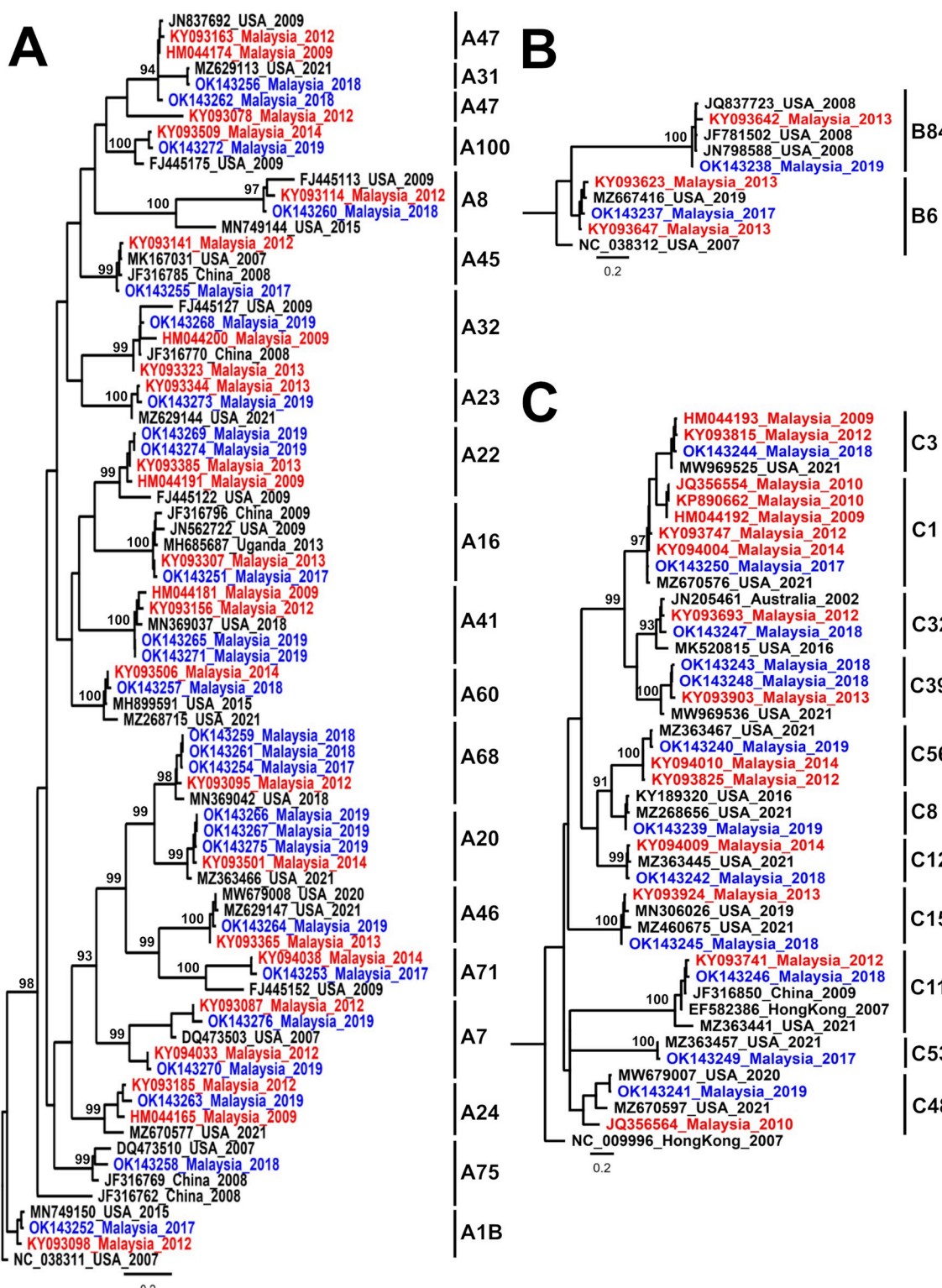

**Fig 2. Phylogenetic trees of rhinovirus A, B and C targeting partial VP4/VP2 gene sequences.** Strain names are in the format: accession number_country of isolation_year of isolation. The numbers refer to percentage of bootstrap support at key nodes. Malaysian sequences are coloured red and sequences from this study are coloured blue. The phylogenetic trees of rhinovirus A, B and C are rooted with reference genomes with accession numbers NC_038311, NC_038312 and NC_009996, respectively.

## Conclusions

In summary, before the pandemic, a high proportion of SARI in adults in Kuala Lumpur, Malaysia, were associated with rhinovirus/enterovirus and influenza virus. Continued surveillance and monitoring of changes in circulating viruses, including emerging pathogens, can contribute to effective prevention strategies. The highly sensitive viral metagenomics approach can identify viral pathogens missed by routine testing and rare or emerging pathogens. However, issues with validation, result interpretation, cost and turnaround time hinder its routine use.

## Supporting information

**S1 Data. Validation of molecular assays.**
(DOCX)

**S1 Table. Primers used for genotyping rhinovirus/enterovirus.**
(DOCX)

## Acknowledgments

We acknowledge BEI Resources, National Institute of Allergy and Infectious Diseases (NIAID), National Institutes of Health (NIH) for providing genomic RNA from MERS-CoV (NR-45843) and gamma-irradiated inactivated SARS-CoV (NR-9547) as standard control for our qPCR assay.

## Author Contributions

**Conceptualization:** Yoke Fun Chan, I-Ching Sam.

**Data curation:** Yoong Min Chong.

**Formal analysis:** Yoong Min Chong, Yoke Fun Chan, I-Ching Sam.

**Funding acquisition:** Yoke Fun Chan, I-Ching Sam.

**Investigation:** Yoong Min Chong, Mohamad Fadhil Hadi Jamaluddin, M. Shahnaz Hasan, Yong Kek Pang, Sasheela Ponnampalavanar, Sharifah Faridah Syed Omar.

**Methodology:** Yoong Min Chong, Yoke Fun Chan, I-Ching Sam.

**Project administration:** Yoke Fun Chan, I-Ching Sam.

**Resources:** Yoke Fun Chan, Mohamad Fadhil Hadi Jamaluddin, M. Shahnaz Hasan, Yong Kek Pang, Sasheela Ponnampalavanar, Sharifah Faridah Syed Omar, I-Ching Sam.

**Supervision:** Yoke Fun Chan, I-Ching Sam.

**Writing – original draft:** Yoong Min Chong, I-Ching Sam.

**Writing – review & editing:** Yoong Min Chong, Yoke Fun Chan, Mohamad Fadhil Hadi Jamaluddin, M. Shahnaz Hasan, Yong Kek Pang, Sasheela Ponnampalavanar, Sharifah Faridah Syed Omar, I-Ching Sam.

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
