## [Decision Letter · Decision Letter 0]

28 Jul 2022

PONE-D-22-15536Rhinovirus/enterovirus was the most common respiratory virus detected in adults with severe acute respiratory infections pre-COVID-19 in Kuala Lumpur, MalaysiaPLOS ONE

Dear Dr. Sam,

Thank you for submitting your manuscript to PLOS ONE. After careful consideration, we feel that it has merit but does not fully meet PLOS ONE’s publication criteria as it currently stands. Therefore, we invite you to submit a revised version of the manuscript that addresses the points raised during the review process.

Reviewers point out the limitations of your research (Reviewer #2). Therefore, I ask the authors to prepare proper responses to these comments and to make appropriate additions to the text of the manuscript.

We look forward to receiving your revised manuscript.

Kind regards,

Ruslan Kalendar

Academic Editor

PLOS ONE

Journal Requirements:

Reviewers' comments:

Reviewer's Responses to Questions

**Comments to the Author**

1. Is the manuscript technically sound, and do the data support the conclusions?

Reviewer #1: Yes

Reviewer #2: Yes

2. Has the statistical analysis been performed appropriately and rigorously? 

Reviewer #1: Yes

Reviewer #2: Yes

3. Have the authors made all data underlying the findings in their manuscript fully available?

Reviewer #1: Yes

Reviewer #2: Yes

4. Is the manuscript presented in an intelligible fashion and written in standard English?

Reviewer #1: Yes

Reviewer #2: Yes

5. Review Comments to the Author

Reviewer #1: 

The aim of the study was to detect viral etiologies in adults admitted with SARI in Kuala Lumpur, Malaysia in the 2 years before COVID-19, using molecular assays and metagenomics analysis. Samples of 489 patients were evaluated by a commercial multiplex nucleic acid assay (Luminex assay). In 57.1% of the patients one or more pathogens were detected. Rhinovirus/enterovirus (RV/EV) was the most prevalent agent that detected in nearly half of the samples, followed by influenza virus. In 105 Luminex negative samples were evaluated by a viral metagenomic analysis. A positive result was detected in 2% samples. In addition, factors related to increased risk of critical disease were studied. RV/EV isolates were characterized by sequencing.

This is a study providing information regarding viral pathogens causing SARI in pre-COVID-19 era in a single center. It also reports the value of further investigation with metagenomics analysis to detect the pathogen in primer assay negative samples. Although results are not unique, the manuscript is well written, methods are clearly described and laboratory results are combined with patients clinical details.

Reviewer #2: 

This manuscript is an epidemiological study investigating the causative virus of adult severe respiratory infections in Malaysia. I don't think there are any particular problems in considering the methods and results of this research.

However, this study is an epidemiological study in a very limited area of Malaysia and is not considered to reflect the whole of Malaysia or Asia.

In addition, this study was performed before the COVID-19 pandemic, and if possible, it would be better to investigate what kind of changes have occurred since the COVID-19 pandemic.

Therefore, I don’t think it will be cited by many people at this point. Or, its scientific value is not high.

6. PLOS authors have the option to publish the peer review history of their article (what does this mean?). If published, this will include your full peer review and any attached files.

Reviewer #1: **Yes: **A Arzu Sayıner

Reviewer #2: No

---

## [Author Response · Author response to Decision Letter 0]

3 Aug 2022

We thank the editor and reviewers for their time and effort. 

As requested, we will first respond to the following prompts listed in the editor's email:

"a) If there are ethical or legal restrictions on sharing a de-identified data set, please explain them in detail (e.g., data contain potentially identifying or sensitive patient information) and who has imposed them (e.g., an ethics committee). Please also provide contact information for a data access committee, ethics committee, or other institutional body to which data requests may be sent."

As our clinical dataset contains at least 3 indirect patient identifiers (as defined by the reference cited by PLoS, i.e. http://www.bmj.com/content/340/bmj.c181.long), that is place of treatment, sex, and age, and there are other potential identifiers such as clinical severity (ICU/death) and year of treatment, our hospital’s Medical Research Ethics Committee felt that there is enough information to potentially identify patients, and therefore this database should not be made publicly available.

Requests for data can be made to:

Chairman,

Medical Research Ethics Committee,

2nd floor, Kompleks Pendidikan Sains Kejururawatan, 

University of Malaya Medical Centre,

Kuala Lumpur 59100, 

Malaysia.

Tel: +603 79493209

E-mail: ummc-mrec@ummc.edu.my

"b) If there are no restrictions, please upload the minimal anonymized data set necessary to replicate your study findings as either Supporting Information files or to a stable, public repository and provide us with the relevant URLs, DOIs, or accession numbers. Please see http://www.bmj.com/content/340/bmj.c181.long for guidelines on how to de-identify and prepare clinical data for publication. For a list of acceptable repositories, please see http://journals.plos.org/plosone/s/data-availability#loc-recommended-repositories."

Not applicable.

Reviewer #1: 

"The aim of the study was to detect viral etiologies in adults admitted with SARI in Kuala Lumpur, Malaysia in the 2 years before COVID-19, using molecular assays and metagenomics analysis. Samples of 489 patients were evaluated by a commercial multiplex nucleic acid assay (Luminex assay). In 57.1% of the patients one or more pathogens were detected. Rhinovirus/enterovirus (RV/EV) was the most prevalent agent that detected in nearly half of the samples, followed by influenza virus. In 105 Luminex negative samples were evaluated by a viral metagenomic analysis. A positive result was detected in 2% samples. In addition, factors related to increased risk of critical disease were studied. RV/EV isolates were characterized by sequencing.

This is a study providing information regarding viral pathogens causing SARI in pre-COVID-19 era in a single center. It also reports the value of further investigation with metagenomics analysis to detect the pathogen in primer assay negative samples. Although results are not unique, the manuscript is well written, methods are clearly described and laboratory results are combined with patients clinical details."

Thank you for your positive review.

Reviewer #2: 

"This manuscript is an epidemiological study investigating the causative virus of adult severe respiratory infections in Malaysia. I don't think there are any particular problems in considering the methods and results of this research.

However, this study is an epidemiological study in a very limited area of Malaysia and is not considered to reflect the whole of Malaysia or Asia."

This is an important limitation which we have acknowledged in the discussion (p23, lines 371-373): 

“It involved a single hospital, and only 105/218 (48.2%) negative samples underwent viral metagenomics analysis. Broader and more extensive surveillance studies are needed for more nationally representative data.”

"In addition, this study was performed before the COVID-19 pandemic, and if possible, it would be better to investigate what kind of changes have occurred since the COVID-19 pandemic."

This is an important point, as other respiratory viruses have been widely shown to have drastically reduced during the pandemic. We have separately published a similar study carried out in the early months of the pandemic which supports this, and have cited this in the discussion of the limitations of our study (p23, lines 375-377):

“This study was conducted before the COVID-19 pandemic, which was associated with declines in other respiratory viruses globally, including at our centre [41,42].”

(Ref 41 is our study, and can be found at https://doi.org/10.1016/j.jcv.2021.105000)

"Therefore, I don’t think it will be cited by many people at this point. Or, its scientific value is not high."

While the data is not novel, we believe, as we have written in the discussion, that “it provides important baseline data of circulating respiratory viruses in a tropical country.” This data is certainly lacking in adults in Malaysia and due to the cost of multiplex PCR/NGS and competing demands on limited diagnostic resources, is unlikely to be widely collected apart from sporadic, well-funded research projects or private hospitals.

Furthermore, it is evident that as COVID-19 numbers have relatively waned this year, there has been a resurgence of respiratory viruses which were much reduced during the pandemic, such as influenza, RSV and adenovirus. Therefore, the post-pandemic virus circulation patterns are more likely to resemble the pre-pandemic circulation of multiple viruses. This adds to the importance of our baseline data for 2017-2019.

---

## [Decision Letter · Decision Letter 1]

15 Aug 2022

Rhinovirus/enterovirus was the most common respiratory virus detected in adults with severe acute respiratory infections pre-COVID-19 in Kuala Lumpur, Malaysia

PONE-D-22-15536R1

Dear Dr. Sam,

We’re pleased to inform you that your manuscript has been judged scientifically suitable for publication and will be formally accepted for publication once it meets all outstanding technical requirements.

Kind regards,

Ruslan Kalendar

Academic Editor

PLOS ONE

---

## [Editor Report · Acceptance letter]

25 Aug 2022

PONE-D-22-15536R1 

Rhinovirus/enterovirus was the most common respiratory virus detected in adults with severe acute respiratory infections pre-COVID-19 in Kuala Lumpur, Malaysia 

Dear Dr. Sam:

I'm pleased to inform you that your manuscript has been deemed suitable for publication in PLOS ONE. Congratulations! Your manuscript is now with our production department. 

Kind regards, 

on behalf of

Professor Ruslan Kalendar 

Academic Editor

PLOS ONE